# Mechanically Tough and Highly Stretchable Hydrogels Based on Polyurethane for Sensitive Strain Sensor

**DOI:** 10.3390/polym15193902

**Published:** 2023-09-27

**Authors:** Jianyang Shi, Shuang Wang, Haibo Wang, Jun Gu

**Affiliations:** 1Department of Cardiovascular Surgery, West China Hospital, Sichuan University, Chengdu 610065, China; sjy812001920@163.com (J.S.); whb6985@scu.edu.cn (H.W.); 2College of Biomass Science and Engineering, Sichuan University, Chengdu 610065, China; shuangshine7@scu.edu.cn

**Keywords:** strain sensor, mechanical tough, highly stretchable, sensitive, MXenes, macromolecular crosslinking agent

## Abstract

Hydrogels with flexible and stretchable properties are ideal for applications in wearable sensors. However, traditional hydrogel-based sensors suffer from high brittleness and low electrical sensitivity. In this case, to solve this dilemma, a macromolecular polyurethane crosslinking agent (PCA) was designed and prepared; after that, PCA and two-dimensional (2D) MXene nanosheets were both introduced into a covalently crosslinked network to enhance the comprehensive mechanical and electrochemical properties of the hydrogels. The macromolecular polyurethane crosslinking agent promotes high-tensile strength and highly stretchable capacity by suitable covalent crosslinking. The optimized hydrogel, which exhibited maximum tensile strength and maximum elongation at break, had results of 1.21 MPa and 644%, respectively. Two-dimensional MXene nanosheets provide hydrogel with high electrical conductivity and strain sensitivity, producing a wearable device for the continuous monitoring of human movements and facial microexpressions. This study demonstrated an efficient structure design strategy for building mechanically tough, highly stretchable, and sensitive dual-mode MXenes-based wearable sensors.

## 1. Introduction

Recently, flexible sensors have been widely used in the field of electronic skin [1,2], wearable electronic devices for healthcare monitoring [3,4], and so on. However, commonly used electronic devices, usually made of carbon-based materials [5,6,7] or conductive polymers [8,9], result in flexible sensors suffering from flexibility and brittleness, which limited their application in human monitoring. Hydrogels, which possess mechanical properties similar to skin, offer a promising solution to this challenge. Recently, hydrogels have been considered the next generation of flexible biosensors due to their desirable flexibility and biocompatibility [10]. Typically, hydrogels form stable three-dimensional structures with the help of a cross-linker. Wang and co-workers [11] reported a sensor based on a PAM/polyaniline composite hydrogel, which could sense strain against human motion. However, the tensile strength of the hydrogel was only 42 kPa. The hydrogel network was cross-linked through small molecules (MBAA), which are the primary causes of the fragility of hydrogels. The poor mechanical properties of hydrogel limit their applications to flexible sensors. Elasticity is important for hydrogels to adapt to tissue movement and deformation.

To improve the mechanical properties of hydrogels, an effective method is to introduce a multifunctional crosslinker into hydrogels, which could improve the mechanical properties. Polyurethane, which consists of flexible soft segments and hard, rigid segments, exhibited excellent mechanical performances due to the energy dissipation mechanism and intermolecular interactions between hard–hard segments as well as hard-soft segments [12,13]. Moreover, polyurethanes, composed of polyisocyanates and macropolyols, are highly versatile because of the range of chemistries that could be employed in their synthesis, resulting in a multitude of structures and properties. Therefore, the introduction of polyurethane into hydrogel could enhance the mechanical properties of flexible sensors.

To further improve the conductivity of flexible sensors, conductive fillers, such as graphene [14,15], metal nanoparticles [16], and conductive polymers, are commonly used in a hydrogel matrix. Among conductive fillers, MXene is considered an attractive candidate for fabricating conductive hydrogels due to its excellent electrical conductivity, and large specific surface area [17,18,19]. With excellent electrical conductivity, MXene could form a continuous conductive network in the hydrogels and exhibit resistance change under external force, realizing the improvement in sensitivity of flexible sensors. Moreover, taking advantage of negatively charged hydrophilic surfaces, the hydrogen bonding interaction between MXene and polymer endows the hydrogel with enhanced mechanical strength. Therefore, using MXene as a conductive filler not only endows flexible sensors with conductivity, but also enhances mechanical properties.

In this case, to solve this dilemma, a macromolecular polyurethane crosslinking agent (PCA) was designed and prepared; after that, PCA and two-dimensional (2D) MXene nanosheets were both introduced into a covalently crosslinked network to enhance the comprehensive mechanical and electrochemical properties of the hydrogels. A macromolecular PCA crosslinking agent promotes high-tensile strength and highly stretchable capacity by suitable covalent crosslinking. The optimized hydrogel, which exhibited maximum tensile strength and maximum elongation at break, showed results of 1.21 MPa and 644%, respectively. Two-dimensional MXene nanosheets endowed hydrogel with high electric conductivity and strain sensitivity, producing a wearable device used for continuous monitoring of human motions and facial micro-expressions. This study demonstrated an efficient structure-design strategy for constructing mechanically tough, highly stretchable, and dual-mode sensitive MXenes-based wearable sensors.

## 2. Materials and Methods

### 2.1. Materials

Ti_3_AlC_2_ powder was provided by Hangzhou Yanqu Co., Ltd. (Hangzhou, China) Polyethylene glycol (PEG 2000), 2, 2-dimethylolpropionic acid (DMPA), hydroxyethyl acrylate (HEA), isophorone diisocyanate (IPDI), and dibutyltin dilaurate (DBTDL) was supplied by Wanhua Chemical Co., Ltd. (Yantai, China). Carboxybetaine methacrylate (CBMA), acrylamide (AM), and solvent butanone in the synthesis process were bought from Aladdin Biochemical Technology Co. (Shanghai, China). Photoinitiator 2-hydroxy-4′-(2-hydroxyethoxy)-2-methyl-propiophe (Irgacure 2959), lithium fluoride, and hydrochloric acid (HCl) were provided by Huaxia Chemical Reagent Co., Ltd. (Chengdu, China). Deionized water was self-made in the laboratory. Bluetooth data transmission device was supplied by LinkZill Technology Co., Ltd. (Hangzhou, China).

### 2.2. Preparation of MXene Nanosheet

The MXene (Ti_3_C_2_T_x_) nanosheets were fabricated by selectively etching out the aluminum layer from the MAX precursor using the LiF/HCl method. Firstly, 40.0 mL HCl (9 M) and 3.2 g LiF powder were added in a Teflon vessel at ambient temperature to prepare the HCl-LiF etchant. Subsequently, 2.0 g Ti_3_AlC_2_ powder was added slowly to the above etchant followed by magnetically stirring at 800 rpm under 40 °C for 48 h. The etched mixture was centrifugated (3500 rpm, 5 min) and the precipitate was washed using DI water repeatedly until the pH ≥ 7. To acquire single layer MXene nanosheets, the washed precipitate was further exfoliated in an aqueous solution by ultrasonicating (100 Hz) in a ice water bath under N_2_ flow for 2 h. Lastly, 30 min centrifugation (3500 rpm) was performed on the above solution. The target MXene product dissolved in the supernatant was collected and sealed in 4 °C for further use and the Mxene content (25.0 mg/mL) was obtained by the freeze-drying method.

### 2.3. Synthesis of Macromolecular Polyurethane Crosslinking Agent (PCA)

Firstly, PEG was dewatered by stirring at 120 °C under reduced pressure for 2 h. Secondly, a small amount of DMPA powder was spread flat on the bottom of a beaker and then placed in a vacuum furnace to dehydrate at 80 °C for 2 h. After that, the IPDI (24.2 g, 109 mmol) and the dried PEG (60.0 g, 30 mmol), DMPA (8.0 g, 60 mmol), and 5 drops of the DBTDL catalyst were added to a 500 mL three-necked, round-bottomed flask equipped with a mechanical stirrer. The mixture was stirred and reacted at 85 °C until the isocyanate group reached the theoretical value; the NCO-terminated group linear oligomer was obtained. Finally, the reaction temperature was reduced to 50 °C with continued stirring, and then HEA (4.10 g, 36 mmol) was added. The reaction was continued for 4 h to obtain a double-bonded, capped, macromolecular polyurethane crosslinking agent (PCA).

### 2.4. Preparation of MXene-Poly(Urethane-Co-AM-Co-CBMA) Hydrogels (Labeled as MPH Nanocomposite Hydrogels)

The MPH nanocomposite hydrogels were prepared as follows: AM (2.0 g), CBMA (64 mg), and PCA were well dispersed in 8.0 g deionized water under magnetic stirring to obtain a homogeneous mixture. After that, the etched 0.2 g of monolayers of MXene and 63 mg of Irgacure 2959 powder were added to the mixture to continue magnetic stirring for adequate dispersion. The mixture was further dispersed by ultrasonication for 5 min and the air bubbles were removed at the same time. The mixture was transferred to a polytetrafluoroethylene mold with a UV lamp and irradiated for 10 min at room temperature to obtain MPH hydrogels (labeled MxPyH, x is the amount of MXene added and y is the mass ratio of PCA to AM). Figure 1 illustrates the synthetic route and preparation of MPH hydrogels. The specific formulation of MPH hydrogel is shown in Table 1. ACH is poly (AM-co-CBMA) hydrogel (without polyurethane cross-linking agent).

### 2.5. Characterization

Fourier Transform Infrared (FT-IR) spectroscopy was performed using a Nicolet 560 FT-IR spectrometer (Thermo Fisher Scientific, Waltham, MA, USA) with a test wavelength range of 4000 to 700 cm^−1^ at a resolution of 4 cm^−1^. Scanning electron microscopy (SEM) analysis was performed using Quanta 250 scanning electron microscopy (FEI, Hillsboro, OH, USA) to analyze the morphology of freeze-dried hydrogel samples. The samples were electrically conductive by spraying gold. The operating voltage of the MXene, ACH, PCA, and MPH hydrogels by scanning electron microscopy was 5 kV.

The electrochemical performance of the assembled hydrogel sensor was analyzed using the ZAHNER ENNIUM electrochemical workstation, and calculate the sensitivity (*S*) of the sensor according to the formula (1)
(1)$$S=\frac{\partial (\Delta R/R_{0}^{})}{\partial P}$$
where Δ*R* = *R_t_* − *R*_0_. *R_t_* and *R*_0_ are the real-time resistance and the initial resistance of the hydrogel, respectively. *P* is the real-time stress or strain of the hydrogel.

The Instron 5967 tensile testing machine (Instron Electron Instrument Co., Ltd., Norwood, MA, USA) was used for the mechanical property testing of hydrogels. Mechanical measurements of hydrogels are performed at a rate of 50 mm/min. The samples are dumbbell-shaped (30 mm length × 4 mm width × 3 mm thickness). In addition, compression tests are performed at 80% using a cylindrical MPH hydrogel with a diameter of 20 mm at a rate of 10 mm/min. To prevent the evaporation of water from the hydrogels, glycerin was used during the tensile test.

## 3. Results

### 3.1. Structural Characterization

The FT-IR curves in Figure 2a record the characteristic peaks of the PCA, ACH, and MPH hydrogels to characterize their chemical structures. The peaks at 1247 cm^−1^, 1541 cm^−1^, 1695 cm^−1^, and 3325 cm^−1^ were attributed to C-O stretching vibration, C-N bending vibration, C=O stretching vibration, and N-H stretching vibration. The appearance of these characteristic peaks is associated with urethane (-NHCOO-) groups. In addition, the spike at 1109 cm^−1^ was derived from C-O-C stretching vibration. Characteristic peaks at 2275 cm^−1^ to 2240 cm^−1^ are associated with isocyanate groups, and their disappearance indicates that isocyanate groups are fully reacted. These results indicate the successful synthesis of PCA. The peaks at 1668 cm^−1^, 3429 cm^−1^, and 1120 cm^−1^ were related to C=O, N-H, -CO-O- stretching vibrations, respectively [20,21]. These peaks appeared in both ACH and MPH. The TEM image depicted the successful preparation of the MXene sheet. In addition, the characteristic peaks of the urethane groups also appeared in MPH, indicating that PCA was successfully introduced into hydrogel. These FT-IR results indicate the successful synthesis of MPH hydrogels.

Cross-sectional microscopic morphologies of lyophilized ACH, PCA, and MPH hydrogels were studied by SEM [22,23,24]. The dense and porous cross-sectional structure of ACH is presented in Figure 3a. This resulted in unsatisfactory mechanical properties of ACH. As shown in Appendix A, it can be observed by the SEM image that the cross-section of PCA shows many folds [25]. As shown in Figure 3b–f, the introduction of PCA into MPH does not disrupt its porous structure. The pore size of MPH is smaller than that of the pure ACH hydrogel, which substantially improves the mechanical properties of MPH compared to pure ACH hydrogel. In addition, a large number of groups on the surface of MXene nanosheets were able to form many hydrogen bonds with MPH, further improving the mechanical properties of MPH. As a result, MPH hydrogels can resist external stretching and compression, solving part of the problem of traditional hydrogels that are easily damaged by external forces [26,27,28]. Figure 3c shows a monolayer of MXene uniformly dispersed in MPH hydrogels. MXene did not show significant aggregation in MPH, indicating that its introduction did not cause serious degradation of hydrogel properties.

### 3.2. Mechanical Properties

The comprehensive mechanical properties of hydrogels are key factors in the preparation of wearable sensors. To expand the application fields of hydrogels and solve the problems of traditional hydrogels, sensors need to have excellent elasticity, mechanical strength, and durability [29,30]. To investigate the effect of different PCA contents on the mechanical properties of MPH hydrogels, a universal material testing machine is carried out to investigate the mechanical properties of hydrogels. Figure 4a exhibited the gradual stretching of MPH from its original length to 300% of its original length. No damage, such as tearing, occurred in the hydrogel during this process. When the external force was removed, it could immediately revert to its initial state, demonstrating that the MPH hydrogel had outstanding self-recovery capabilities and good mechanical properties [31,32,33].

As shown in Figure 4b, the tensile strength of MPH hydrogels increased gradually as PCA content increased from 0 wt% to 0.5 wt%. The introduction of double-bond capped polyurethane as a crosslinking agent in MPH hydrogels markedly improved the tensile strength of the hydrogels [34]. As the content of PCA increased to 1 wt%, the crosslink density increased. However, MPH’s tensile strength began to decline. In addition, there was a substantial decrease in strength and elongation at break when PCA content increased to 2 wt%. It can be explained by the fact that excessive cross-linking causes difficulties in the movement of molecular chains, and smaller forces cause them to break. This also explains the gradual decrease in elongation at break due to the decrease in chain movement with increasing PCA content in Figure 4c. As a result, the high-PCA content caused excessive cross-linking, which rendered the MPH hydrogel brittle and showed a substantial decrease in both elongation at break and strength [35,36,37]. The Young’s modulus of ACH, M_0.2_P_2_H, M_0.2_P_1_H, M_0.2_P_0.5_H, M_0.2_P_0.25_H, and M_0.2_P_0_H were 0.35, 1.19, 1.82, 1.85, 1.75, and 0.98 MPa, respectively. With the addition of MXene, the Young’s modulus of the hydrogel was significantly improved. With the addition of PCA, the Young’s modulus of the hydrogel first rose to a maximum of 1.85 MPa and then decreased, which proved that the increase in crosslinking degree would significantly improve the hydrogel’s strength, but the excess degree of crosslinking would hinder the movement of the polymer chain and reduce the hydrogel’s stiffness. Therefore, there is an optimal choice for the introduced content of PCA to balance the flexibility and toughness of the MPH hydrogel. Through this series of mechanical property tests, when the PCA content is 0.5 wt%, the comprehensive mechanical properties of the MPH hydrogel are the best, so the M_0.2_P_0.5_H hydrogel is used for the next research test.

The use of hydrogels is inevitably confronted with repetitive deformation processes over a long period, so excellent fatigue resistance and recoverability are essential [38]. During cyclic stretches, the strain gradually increased from 100% to 400%, and the M_0.2_P_0.5_H hydrogel showed a small hysteresis (Figure 5a) with excellent stability. According to Figure 5b, MPH hydrogels have an excellent linear relationship between stress and strain. The introduction of a PCA crosslinking agent could form covalent crosslinking with the ACH hydrogel. In addition, the large number of polar groups on PCA can form abundant hydrogen bonds with the MPH molecular chain, thus greatly improving the stability of the hydrogel’s 3D-crosslinked network [39]. In summary, the dual crosslinking formed by PCA can significantly improve the comprehensive mechanical properties of MPH hydrogels.

### 3.3. Electrical and Sensing Properties

A dual-mode sensor based on the M_0.2_P_0.5_H hydrogel was assembled and data collected as shown in Figure 6a. When the M_0.2_P_0.5_H hydrogel was employed as a stress sensor, sensitivity *S*_1_ of the sensor was calculated as 10.09 kPa^−1^ in the stress range of 0–6 kPa. However, when the stress was in the range of 6 kPa to 15 kPa, the sensitivity *S*_2_ was calculated as 1.76 kPa^−1^. Although its sensitivity decreases at higher pressures, the signal remains stable. As a motion monitoring sensor, its strain sensing capability is also essential. As shown in Figure 6b, its tensile sensitivity was investigated in the strain range of 0–200%. Based on the relative resistance change curve of the M_0.2_P_0.5_H hydrogel as a strain sensor, its sensitivity *S*_1_ was calculated to be 1.17 at 0–100% strain. As the strain continued to increase, the sensitivity of the M_0.2_P_0.5_H hydrogel at 100–200% strain increased substantially, and the sensitivity of *S*_2_ reached 2.62.

It is important to study the stability of the relative resistance change signal of hydrogels as sensors during their long-term use. The strain sensor was tested for 1000 tensile cycles. In the experimental durability test shown in Figure 7a, the M_0.2_P_0.5_H hydrogel was subjected to 1000 tensile cycles at 50% strain. Figure 7b–d further amplifies the relative resistance change curves for the first, middle, and last 10 cycles of the hydrogel tensile cycles. The results showed that the relative resistance change of the hydrogel’s sensors in cyclic tensile tests was almost unchanged from cycle to cycle [40,41]. The above results suggest that the M_0.2_P_0.5_H hydrogel sensor has excellent electrochemical stability and good fatigue resistance.

### 3.4. Human Motion Monitoring Application

As shown in Figure 8a, the M_0.2_P_0.5_H hydrogel was assembled as a strain sensor using copper wires as conductive wires to monitor various behaviors of the human body. Additionally, connecting sensors and phones via Bluetooth for data collection and processing. The MPH hydrogel undergoes a corresponding change in strain with body movement, resulting in a change in electrical resistance. At the same time, a Bluetooth data transmitter collects the signal of the resistance change of the M_0.2_P_0.5_H and transmits it to the smartphone [42]. Finally, the smartphone processes the resistance change data to draw a relative resistance change curve, thus achieving the purpose of real-time monitoring of human movement.

As shown in Figure 8b, when the finger was straightened to present 0°, the MPH hydrogel was bonded to the finger. In this way, the relationship between the degree of finger bending and the relative resistance change of the M_0.2_P_0.5_H hydrogel is tracked. According to the relative resistance change curve, the relative resistance change gradually increases as the finger is slowly bent from 0° to 90° [43,44]. The signal change at the same angle is small, indicating that the signal is stable every time the finger bends. Therefore, it can accurately recognize the bending angle of the finger by the magnitude of the relative resistance change during use.

Research on the M_0.2_P_0.5_H hydrogel sensor is displayed in Figure 8c–g to examine its applicability to tracking human movement. Based on the relative resistance change curves of the human body during movements of the abdomen (Figure 8f) and wrist (Figure 8d), it is shown that the MPH hydrogel sensor is capable of monitoring small-amplitude human movement changes. Although the changes are small, all relative resistance change curves are clear and stable [45,46,47]. As a human motion monitoring sensor, large-amplitude human joint movement monitoring is also important. The relative resistance curves of human elbows (Figure 8c), ankles (Figure 8e), and knees (Figure 8g) were recorded. As with small-amplitude human movements, the MPH hydrogel was able to obtain clear and stable signals of relative resistance changes for large-amplitude human motions [48]. It suggests that, due to the excellent mechanical properties and sensitivity of M_0.2_P_0.5_H hydrogel under large strains, it can monitor large-amplitude human motion changes without signal loss and failure. Based on these results, the M_0.2_P_0.5_H hydrogel has excellent monitoring capabilities for both larger and smaller human movements. This allows the M_0.2_P_0.5_H hydrogel to monitor the movement of most joints in the human body, which greatly broadens the application field of the MPH hydrogel [49,50].

## 4. Conclusions

In summary, an MXene nanosheet layer was first obtained by etching Ti_3_AlC_2_ through the LiF/HCl etching method. Then, IPDI, PEG, and DMPA are used to synthesize the prepolymer, which is then capped by HEA to obtain a polyurethane crosslinking agent with double-bond capping. After adding MXene and mixing well, PCA was used as a cross-linking reagent, and AM and CBMA were copolymerized as monomers under UV irradiation. An MXene-doped poly(Urethane-co-AM-co-CBMA) copolymerization MPH hydrogel was prepared. The chemical structures of pure ACH, PCA, and MPH hydrogels were characterized by FT-IR. The microscopic morphologies of the ACH, PCA, and MPH hydrogel were observed and analyzed by SEM. The mechanical properties were investigated at different PCA contents, and M_0.2_P_0.5_H was selected as the hydrogel with comprehensive performance for further sensing performance tests. Finally, the M_0.2_P_0.5_H was combined with a Bluetooth transmission device to form a human motion monitoring sensor that clearly and stably monitors human joint movements at different amplitudes. It has great potential for its application to human motion monitoring.

## Figures and Tables

**Figure 1 polymers-15-03902-f001:**
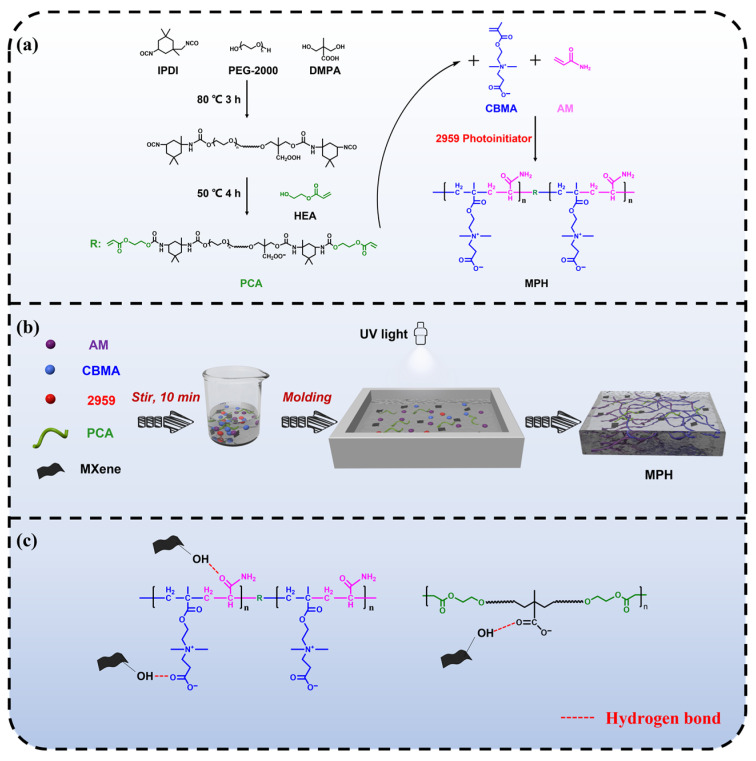
Schematic illustration of the preparation of the MPH nanocomposite hydrogels. (**a**) Synthesis route of PCA and hydrogel, (**b**) preparation process of MPH, (**c**) schematic representation of the interaction of MXene with MPH.

**Figure 2 polymers-15-03902-f002:**
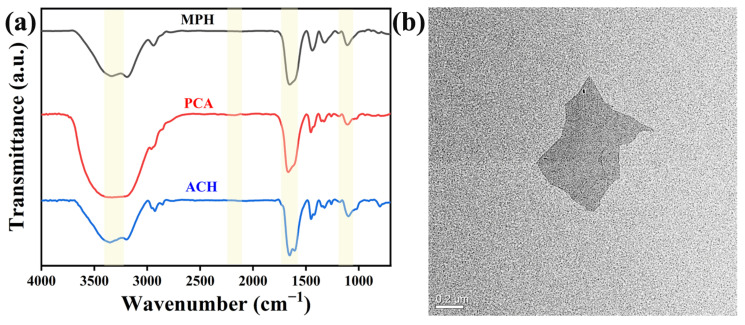
(**a**) FT-IR spectra of ACH, PCA, and MPH hydrogels; (**b**) TEM image of MXene nanosheet.

**Figure 3 polymers-15-03902-f003:**
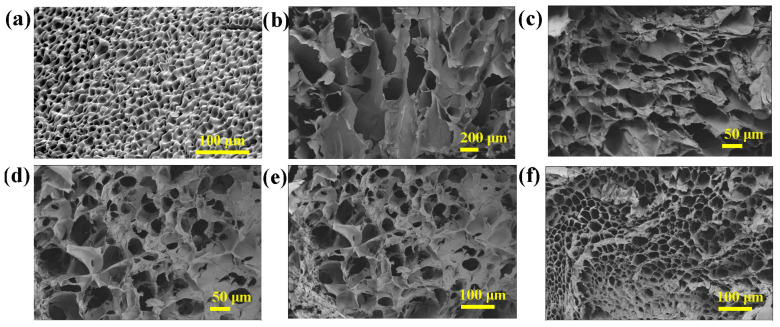
Cross-sectional SEM images of lyophilized (**a**) ACH hydrogel, (**b**–**f**) M_0.2_P_x_H hydrogel.

**Figure 4 polymers-15-03902-f004:**
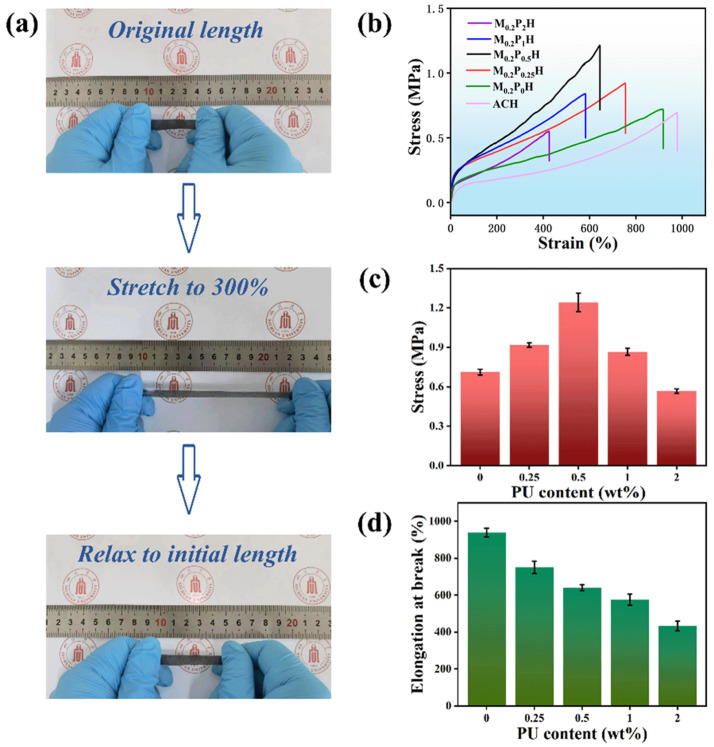
Stretchability of MPH hydrogels. (**a**) Photographs of MPH hydrogel at original length, stretched to 300% and relaxed to original length; (**b**) Typical stress–strain curves of the MPH hydrogel; (**c**) Elongation at break and (**d**) tensile strength of MPH hydrogels with different PCA contents.

**Figure 5 polymers-15-03902-f005:**
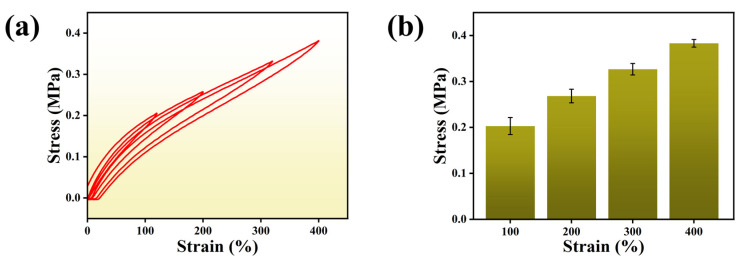
Mechanical properties of the M_0.2_P_0.5_H hydrogel. (**a**) The cyclic tensile curve of the M_0.2_P_0.5_H hydrogel from 100% to 400% strain; (**b**) Tensile stresses corresponding to the M_0.2_P_0.5_H hydrogel at different tensile strains.

**Figure 6 polymers-15-03902-f006:**
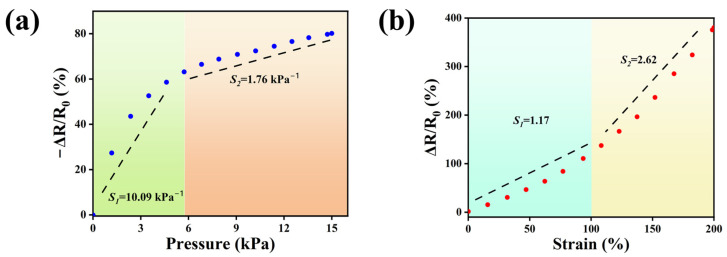
The relative resistance changes of the M_0.2_P_0.5_H hydrogel as (**a**) the stress sensor and (**b**) the strain sensor.

**Figure 7 polymers-15-03902-f007:**
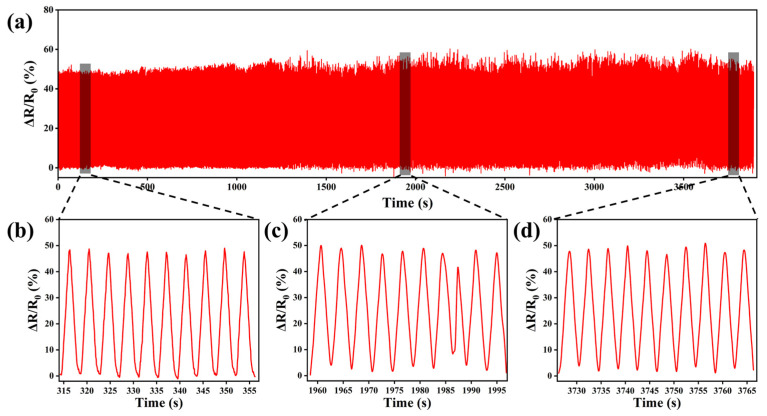
The sensing performance of the M_0.2_P_0.5_H hydrogel. (**a**) The relative resistance changes of the M_0.2_P_0.5_H hydrogel sensor in 1000 cycles of stretching; (**b**–**d**) The relative resistance changes corresponding to each period.

**Figure 8 polymers-15-03902-f008:**
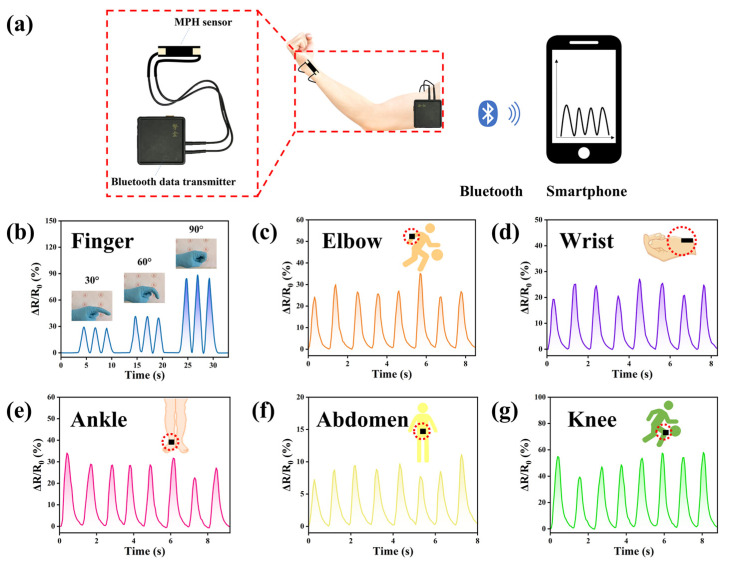
(**a**) Schematic diagram of the M_0.2_P_0.5_H hydrogel as a wireless sensor (Insert Zhihe Bluetooth receiver); the M_0.2_P_0.5_H hydrogel was used to monitor changes in the relative resistance of various parts of the body, such as (**b**) finger, (**c**) elbow, (**d**) wrist, (**e**) ankle, (**f**) abdomen and (**g**) knee.

**Table 1 polymers-15-03902-t001:** The chemical composition of MPH hydrogels.

Sample	PCA (wt%)	AM (g)	CBMA (mg)	2959 (mg)	MXene (g)
M_0.2_P_0_H	0	2.0	64	63	0.20
M_0.2_P_0.5_H	0.5	2.0	64	63	0.20
M_0.2_P_1_H	1.0	2.0	64	63	0.20
M_0.2_P_2_H	2.0	2.0	64	63	0.20
M_0.2_P_4_H	4.0	2.0	64	63	0.20

## Data Availability

Not applicable.

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
