# Peer review of "Mechanically Tough and Highly Stretchable Hydrogels Based on Polyurethane for Sensitive Strain Sensor"

_polymers, 2023, doi:10.3390/polym15193902_

Round 1

Reviewer 1 Report

In the manuscript by Shi et al., the authors presented the design of mechanically robust hydrogels with a polyurethane crosslinker. The introduction of MXene nanosheets into the hydrogels further enhanced the mechanical strength and electrical conductivity. The hydrogels were subsequently fabricated into strain sensors, which were found to have good responsive sensitivity. The presentation of the manuscript was well designed, with schematics and figures carefully organized. However, I found a few major scientific flaws, some of which are important arguments that will undermine the significance of this work. Therefore, I cannot recommend the publication of this manuscript before those concerns have been appropriately addressed. The detailed comments are as follows:

1.     The authors claim that the introduction of the polyurethane crosslinker into the hydrogel would enhance the mechanical properties of the hydrogel. However, from Figure 4, it is unclear how significant the presence of the polyurethane crosslinker is in improving the mechanical performance. The Young's modulus of the samples appears to be constant with varying crosslinking content and the stretchability deteriorates with higher crosslinking. It is unclear whether the polyurethane crosslinker is beneficial.

2.     The authors also claimed that MXene, due to the non-covalent interactions with the polymer network, would further enhance the mechanical properties. This is, however, unclear without examining a control hydrogel in the absence of MXene. When performing this control experiment, it's important to note that the hydrogels with and without MXene should have the same crosslinking density. Given that the authors were using Irgacure 2959 as the initiator and MXene has UV absorption, the degree of polymerization has to be carefully controlled to achieve the same crosslinking density.

3.     The FT-IR spectra of the crosslinker and the hydrogels in Figure 2 might have been mis-labeled. The black trace might be the crosslinker whereas the other two are the hydrogels. It would be necessary to double check. In addition to that, if the FT-IR spectra were directly taken on the hydrogel samples, the peaks at ~1600 and 3300 wavenumbers are largely attributed to water peaks. It is incorrect to assign them as what the authors suggested as they were mostly buried in the water peak.

4.     The acronym PCA was not defined and the NMR spectra (1H and 13C) of the crosslinker in each step of synthesis should be provided with their yields properly reported.

Author Response

Dear  Reviewers:

Thank you for your letter and for the reviewers’ valuable comments concerning our manuscript polymers-2595299 entitled "Mechanically tough and highly stretchable hydrogels based on polyurethane for sensitive strain sensor". We have addressed the comments raised by the reviewers, and the amendments are highlighted in red in the revised manuscript. Point by point response to the reviewers’ comments are listed below this letter. We wish this revised manuscript could be meeting with approval.

Yours sincerely,

Responds to the Reviewer:

Referee: 1

Comments to the Author

In the manuscript by Shi et al., the authors presented the design of mechanically robust hydrogels with a polyurethane crosslinker. The introduction of MXene nanosheets into the hydrogels further enhanced the mechanical strength and electrical conductivity. The hydrogels were subsequently fabricated into strain sensors, which were found to have good responsive sensitivity. The presentation of the manuscript was well designed, with schematics and figures carefully organized. However, I found a few major scientific flaws, some of which are important arguments that will undermine the significance of this work. Therefore, I cannot recommend the publication of this manuscript before those concerns have been appropriately addressed. The detailed comments are as follows:

  1. The authors claim that the introduction of the polyurethane crosslinker into the hydrogel would enhance the mechanical properties of the hydrogel. However, from Figure 4, it is unclear how significant the presence of the polyurethane crosslinker is in improving the mechanical performance. The Young's modulus of the samples appears to be constant with varying crosslinking content and the stretchability deteriorates with higher crosslinking. It is unclear whether the polyurethane crosslinker is beneficial.

Response: Thank you for your valuable comment. As calculated, the Young's modulus of ACH, M0.2P2H, M0.2P1H, M0.2P0.5H, M0.2P0.25H, and M0.2P0H were 0.35, 1.19, 1.82, 1.85, 1.75, and 0.98 MJ/m3, respectively. With the addition of MXene, Young's modulus of the hydrogel was significantly improved. With the addition of PCA, Young's modulus of the hydrogel first rose to a maximum of 1.85 MJ/m3 and then decreased, which proved that the increase of the degree of cross-linking would significantly improve the strength of the hydrogel, but the excess degree of cross-linking would hinder the movement of the polymer chain and reduce the rigidity of the hydrogel. Therefore, it is clear that the polyurethane crosslinker is beneficial.

  1. The authors also claimed that MXene, due to the non-covalent interactions with the polymer network, would further enhance the mechanical properties. This is, however, unclear without examining a control hydrogel in the absence of MXene. When performing this control experiment, it's important to note that the hydrogels with and without MXene should have the same crosslinking density. Given that the authors were using Irgacure 2959 as the initiator and MXene has UV absorption, the degree of polymerization has to be carefully controlled to achieve the same crosslinking density.

Response: Thank you for your valuable comment. We have added the tensile-strain curve of ACH hydrogel in the revised manuscript.

  1. The FT-IR spectra of the crosslinker and the hydrogels in Figure 2 might have been mis-labeled. The black trace might be the crosslinker whereas the other two are the hydrogels. It would be necessary to double check. In addition to that, if the FT-IR spectra were directly taken on the hydrogel samples, the peaks at ~1600 and 3300 wavenumbers are largely attributed to water peaks. It is incorrect to assign them as what the authors suggested as they were mostly buried in the water peak.

Response: Thank you for your valuable comment. We have revised the FT-IR spectra of the crosslinker and the hydrogels in Figure 2.

  1. The acronym PCA was not defined and the NMR spectra (1H and 13C) of the crosslinker in each step of synthesis should be provided with their yields properly reported.

Response: Thank you for your valuable comment. We have defined the acronym PCA (polyurethane crosslinking agent) in Abstract. And the 1H NMR spectra of the crosslinker was provided with their yields (90%) in the revised manuscript.

Reviewer 2 Report

In this study, MXene was introduced into PU-based hydrogel via in situ, photoinitiated polymerization to not only enhance the mechanical property but also endow the electric conductivity. The as-synthesized hydrogel M0.2P0.5H hydrogel shows a strain sensitivity. It is an interesting work. Here are a few comments.

1.        What is ACH in Figure 2? If the disappearances of 2275-2240 cm-1 indicates that the isocyanate groups are fully reacted as the authors claimed, it is confusing that MPH has strong absorption within this wavelength. FTIR analysis should be applied to the hydrogels with different compositions.

2.        The authors should provide SEM images for all 5 MPH groups so as to explain the effect of various compositions. SEM images of MXene need to be obtained.

3.        Please improve the resolution of Figure 4. The author should explain the inverse trend of Figure 4c when PU content is lower or higher than 0.5 wt%.

Major revision by a native English-speaking scientist is mandatory.

Author Response

List of Responses

Dear Reviewers:

Thank you for your letter and for the reviewers’ valuable comments concerning our manuscript polymers-2595299 entitled "Mechanically tough and highly stretchable hydrogels based on polyurethane for sensitive strain sensor". We have addressed the comments raised by the reviewers, and the amendments are highlighted in red in the revised manuscript. Point by point response to the reviewers’ comments are listed below this letter. We wish this revised manuscript could be meeting with approval.

Yours sincerely,

Responds to the Reviewers:

Referee:

Comments to the Author

In this study, MXene was introduced into PU-based hydrogel via in situ, photoinitiated polymerization to not only enhance the mechanical property but also endow the electric conductivity. The as-synthesized hydrogel M0.2P0.5H hydrogel shows a strain sensitivity. It is an interesting work. Here are a few comments.

  1. What is ACH in Figure 2? If the disappearances of 2275-2240 cm-1 indicates that the isocyanate groups are fully reacted as the authors claimed, it is confusing that MPH has strong absorption within this wavelength. FTIR analysis should be applied to the hydrogels with different compositions.

Response: Thank you for your valuable comment. ACH is poly (AM-co-CBMA) hydrogel (without polyurethane cross-linking agent). And we have added it in the revised manuscript. Also, we have revised FT-IR spectra in the revised manuscript.

  1. The authors should provide SEM images for all 5 MPH groups so as to explain the effect of various compositions. SEM images of MXene need to be obtained.

Response: Thank you for your valuable comment. Fig. 2b TEM images of MXene nanosheet.

  1. Please improve the resolution of Figure 4. The author should explain the inverse trend of Figure 4c when PU content is lower or higher than 0.5 wt%.

Response: Thank you for your valuable comment. We have revised the resolution of Figure 4. And we have explained the inverse trend of Figure 4c when PU content is lower or higher than 0.5 wt% in the revised manuscript.

Round 2

Reviewer 1 Report

I much appreciate the authors' efforts in revising the manuscript. It is clear that the manuscript has been greatly improved with all the revisions. However, before this manuscript can be accepted, one minor issue needs to be addressed. The authors described Young's modulus variations of the samples when changing the crosslinker density, but the unit for Young's modulus was MJ/m3. The authors need to check if this is Young's modulus or the work of rupture, the former has the unit of kPa but the latter with the unit of MJ/m3.

Author Response

Comments to the Author

I much appreciate the authors' efforts in revising the manuscript. It is clear that the manuscript has been greatly improved with all the revisions. However, before this manuscript can be accepted, one minor issue needs to be addressed. The authors described Young's modulus variations of the samples when changing the crosslinker density, but the unit for Young's modulus was MJ/m3. The authors need to check if this is Young's modulus or the work of rupture, the former has the unit of kPa but the latter with the unit of MJ/m3.

Response: Thank you for your valuable comment. We have used the unit of MPa for Young's modulus to replace the unit of MJ/m3 in the revised manuscript.

Reviewer 2 Report

All issue of mine were clearly addressed.

Minor editing of English language required

Author Response

Comments to the Author

Minor editing of English language required.

Response: Thank you for your valuable comment. English language and grammar have been carefully revised in the revised manuscript.
